# A U.S. Government-Coordinated Effort to Leverage Non-Human Primate Data to Facilitate Ebolavirus Vaccine Development

**DOI:** 10.3390/vaccines10081201

**Published:** 2022-07-28

**Authors:** Kimberly L. Taylor, Lynda Lanning, Lawrence Wolfraim, Sonia Shrivastava Gales, Colleen Sico, William E. Dowling, Lucy A. Ward, William C. Florence, Edwin Nuzum, Paula R. Bryant

**Affiliations:** 1National Institute of Allergy and Infectious Diseases, National Institutes of Health, Bethesda, MD 20892, USA; larry.wolfraim@nih.gov (L.W.); sonia.gales@nih.gov (S.S.G.); colleen.sico@nih.gov (C.S.); clint.florence@nih.gov (W.C.F.); edn4020@gmail.com (E.N.); paula.bryant@nih.gov (P.R.B.); 2Center for Drug Evaluation and Research, U.S. Food and Drug Administration, Silver Spring, MD 20993, USA; lynda.lanning@fda.hhs.gov; 3Coalition for Epidemic Preparedness Innovations, Washington, DC 20006, USA; william.dowling@cepi.net; 4U.S. Department of Defense (DOD), Joint Program Executive Office for Chemical, Biological, Radiological and Nuclear Defense (JPEO-CBRND), Joint Project Manager for Chemical, Biological, Radiological and Nuclear Medical (JPM CBRN Medical), Fort Detrick, MD 21702, USA; lucy.a.ward.civ@army.mil

**Keywords:** vaccine development, filovirus, ebolavirus, non-human primate, public–private partnership, regulatory, Animal Rule, FANG

## Abstract

A United States Government (USG) interagency group, the Filovirus Animal Non-Clinical Group (FANG), has been established to support the development of biodefense medical countermeasures (MCMs). As both vaccines and therapeutics are licensed using “non-traditional pathways”, such as the U.S. Food and Drug Administration’s (FDA) Animal Rule (AR), non-human primate (NHP) models and associated assays have been developed and standardized across BSL4 testing sites to evaluate candidate products. Vaccine candidates are evaluated using these NHP models, and through this public–private partnership, a meta-analysis of NHP control data has been conducted and submitted to the FDA as a master file. This is an example of how existing NHP control data can be leveraged in lieu of conducting separate natural history studies at multiple testing facilities to demonstrate the consistency of a standardized animal model for vaccine development. As a result, animal use can be minimized and the duplication of effort avoided, thus reducing the amount of time needed to conduct additional studies, as well as the cost of vaccine candidate development. This successful strategy may be applied to other pathogens of high consequence for vaccine development, and shows how strategic preparedness for biodefense can be leveraged in response to outbreaks and public health emergencies.

## 1. Introduction

Filoviruses were discovered in 1967 and have since been responsible for over 40 disease outbreaks, the majority of which were caused by an Ebola virus (as opposed to a Marburg virus), including the devastating West African outbreak from 2013 to 2016 that affected the countries of Guinea, Sierra Leone, and Liberia [1]. Prior to West Africa’s outbreak, filovirus-associated disease outbreaks were sporadic and largely confined to limited numbers of people localized in rural or remote areas of central Africa, thus dampening the pharmaceutical industry’s interest in developing a vaccine. Although there was no available licensed vaccine for Ebola virus disease (EVD) in 2013, there was nonetheless a somewhat “robust” pipeline of vaccine candidates with demonstrated preclinical efficacy (and at least one with phase I clinical safety data). This portfolio was the result of decades of research fueled largely by government funding dedicated to developing MCMs for high-consequence pathogens of biodefense concern. A collaborative effort coordinated by the U.S. government to support vaccine development is the focus of this review. 

## 2. Clinical Development of Vaccines and Regulatory Pathways to Licensure

The West African outbreak accelerated the advancement of several of these vaccine candidates, ultimately resulting in the recent licensure of two Ebola virus vaccines: ERVEBO^®^ (Merck, Kenilworth, NJ, USA) in the U.S. and the European Union (EU), and the Zabdeno/Mvabea vaccine regimen (Janssen, Pharmaceutical Companies of Johnson & Johnson, New Brunswick, NJ, USA) in the EU. Although both of these vaccines entered clinical development during the West African outbreak, only ERVEBO^®^ was able to generate human immunogenicity and safety data in time to enter a 2015 phase III ring vaccination trial in Guinea, and ultimately obtain human effectiveness and efficacy data which permitted licensure via the FDA’s accelerated approval pathway [2]. As a result, the ERVEBO^®^ and Zabdeno/Mvabea vaccines followed different pathways to licensure, and their lessons learned are important to consider for the development of the Sudan and Marburg vaccines.

Both ERVEBO^®^ and Zabdeno/Mvabea vaccines are viral vectored vaccines that have demonstrated a high level of protection in the standardized NHP model following the intramuscular Ebola Zaire challenge [3,4,5]. ERVEBO^®^ is a live-attenuated, replication-competent, recombinant vesicular stomatitis virus-based vaccine expressing the full-length glycoprotein (GP) of a Zaire Ebolavirus (rVSV-ZEBOV), instead of its native VSV GP [6,7]. Results from the 2015 phase III ring vaccination study in Guinea indicated that ERVEBO^®^ was 100% effective, as no new EVD cases were identified 10 or more days following the immunization of vaccinated close contacts [2]. These results were confirmed when ERVEBO^®^ was deployed in the second largest Ebola outbreak ever recorded, between 2018 and 2020 in the Democratic Republic of the Congo; it was shown to be 97.5% effective at stopping Zaire ebolavirus Kikwit (EBOV) transmission among vaccinees [8]. 

In 2015, the Zabdeno/Mvabea vaccine was not yet positioned for evaluation in a phase III efficacy study and instead entered a safety and immunogenicity clinical trial in Sierra Leone. The Zabdeno/Mvabea vaccine utilizes Janssen/J&J’s AdVac platform and Bavarian Nordic’s MVA-BN technology in a heterologous prime boost strategy [5]. The Zabdeno (Ad26.ZEBOV-GP) component serves as the prime and is composed of adenovirus serotype 26 expressing the ZEBOV full-length GP in place of the replication-essential adenovirus early 1 region, rendering it replication incompetent. Mvabea (MVA-BN-filo) serves as the boost and consists of a modified Vaccinia Ankara virus (MVA) encoding GPs from EBOV, Sudan virus (SUDV), and Marburg virus (MARV), and Taï Forest virus (TAFV) nucleoproteins. As the number of cases started to decline in the West African outbreak, an evaluation of clinical efficacy was not possible for this vaccine. Instead, the company continued to develop Zavdeno/Mvabea through the conduct of multiple phase I/II/III clinical trials, demonstrating its safety and immunogenicity, along with additional efficacy studies in the NHP model, and pursued a “non-traditional pathway” to licensure [9,10,11,12]. In 2019, the Committee for Medicinal Products for Human Use (CHMP) of the European Medicines Agency (EMA) granted an accelerated assessment to Janssen for its Ebola vaccine, and Janssen submitted two Marketing Authorization Applications (MAAs) to the EMA for the approval of Ad26.ZEBOV and MVA-BN-Filo. The evaluation of the protective effect of the vaccine regimen was demonstrated through the bridging of clinical immunogenicity results to efficacy and immunogenicity data obtained in non-human primates (NHPs) [13]. In May 2020, the EMA’s CHMP recommended granting a marketing authorization for the combination of Ad26.ZEBOV (Zabdeno) and MVA-BN-Filo (Mvabea) vaccines. 

Indeed, prior to the West African Ebola outbreak, it was assumed that the licensure of MCMs for filoviruses would not be feasible through a traditional clinical efficacy trial; therefore, regulatory agencies established guidelines for demonstrating a likelihood of clinical benefit that could facilitate a non-traditional path to licensure, such as the EMA’s approval under exceptional circumstances that was used for the Zabdeno/Mvabea vaccine as described above. In the U.S., the FDA established the Animal Rule (AR) regulatory pathway for the approval of drug and biological products for which human efficacy studies are not ethical or feasible (described in 21 CFR 314.600–650 for drugs and 21 CFR 601.90–95 for biologics; effective 1 July 2002) [14]. 

Under the FDA’s AR, efficacy is established based on studies in animal models of human disease or conditions of interest using a relatively small number of animal subjects compared to the large-scale phase II or III efficacy trials in humans required to support the licensure of most new drugs and vaccine candidates. As the FDA relies on these studies to demonstrate the primary evidence of product effectiveness, studies conducted must be adequately designed and well-controlled to ensure the rigor of study control, data reliability and data integrity. As such, the animal model used for efficacy studies should be well-characterized and share in-common measurable disease endpoints with those found in humans. Typically, natural history studies that define the animal model are conducted at each testing facility, and when the animal model is shown to be suitable, the data packages are submitted to the FDA in support of the animal model use under the AR regulatory pathway. However, adequate and well-controlled natural history studies for Zaire ebolavirus were limited at the Animal Biosafety Level 4 (ABSL4) sites. 

## 3. U.S. Government-Coordinated Effort to Develop Standardized Tools for Vaccine Development

To coordinate the development of biodefense MCMs, a USG interagency group, the Filovirus Animal Non-Clinical Group (FANG), was established in 2011 to support the development of vaccines and therapeutics to be licensed using non-traditional pathways such as the FDA’s AR [15]. The FANG was instrumental in establishing standardized tools, such as animal models, associated viral and immunogenicity assays, and reagents to support product development prior to the West African outbreak. As these tools were already in place, product developers were able to utilize these tools and capabilities to facilitate the evaluation of candidate products and more rapidly provide data for regulatory submissions to the EMA and FDA for the licensure of products. 

The FANG Animal Models Subteam, which included USG sponsors, as well as academic, private, and industry partners, collaborated to develop NHP models suitable for the evaluation of vaccine candidates and therapeutics under development using the FDA AR regulatory pathway. Through this public–private partnership, the National Institute of Allergy and Infectious Diseases (NIAID), the Department of Defense (DOD), and the Biomedical Advanced Research and Development Authority (BARDA) have sponsored numerous studies to develop and refine animal models, as well as screen and evaluate vaccine candidates. The NHP efficacy study parameters, challenge materials and dosing, disease endpoints and measures, and the associated viral and immunological assays used in the NHP studies were harmonized to the extent possible through the FANG government stakeholders in collaboration with the ABSL4 testing facilities and product developers.

The cynomolgus macaque (CM) model has been a commonly used model for the evaluation of Ebola vaccine candidates, as CMs are susceptible to infection and disease caused by the wild-type ebolavirus [16,17,18], and subsequently it has emerged as a lead NHP model to support vaccine development under the FDA’s AR regulatory pathway. Upon challenge, Ebola virus causes a disease in the CM which closely aligns with the human disease pathogenesis, including clinical signs, clinical chemistry and hematology values, and ultimately mortality. However, the CM model is considered very stringent as it frequently requires a very high dose (100–1000 PFU) of EBOV for intramuscular challenge, and the progression of the disease course is shortened in CMs as compared to human disease. The CM model is also typically uniformly lethal, whereas in humans, the case fatality rate varies from 25% to 90% [19]. Access to the CM model is limited, as it requires the work be performed in a high containment lab (ABSL4), which is a bottleneck for multiple reasons, including the high cost of performing studies in the ABSL4, the required technical and scientific expertise, the limited number of sites able to conduct this work, limited capacity at each site, and the high demand or competition for this space. Large licensure-enabling vaccine efficacy studies at a single testing facility are often logistically challenging, and thus, multiple testing facilities would be required to provide an adequate dataset for immunobridging NHP protection to human immunogenicity data. Therefore, to support licensure through the AR regulatory pathway, natural history studies are required at each testing facility to demonstrate the consistency of the animal model across facilities. 

To accumulate an adequately powered natural history dataset to support the CM model for use in evaluating the efficacy of vaccine candidates, the USG sponsors leveraged datasets from challenged, untreated animals (control animals) from multiple immunogenicity and efficacy studies that were conducted at three ABSL4 testing facilities. These studies were performed under quality systems which were well-controlled, with the intent of potential use in regulatory submissions under the AR. USG sponsors with access to datasets from multiple studies conducted a meta-analysis of the combined datasets to describe the course of Ebolavirus disease in the CM model (natural history). Details of the CM animal model control meta-analysis are described in Niemuth, et al., 2021 [20], and the project was funded through the NIAID Preclinical Services Contract. A government interagency working group, led by NIAID Program scientists, was convened to refine the approach and statistical analysis plan, and to review the final report. Coordination and collaboration among government partners (NIAID, DOD, BARDA, FDA), ABSL4 testing facilities, and product sponsors was critical to the success of the project. 

Briefly, the final analysis included data from 122 CM control animals from 33 studies. Although the majority of these studies were funded by the NIAID and DOD, data were also provided by two product developers (Janssen and Bavarian Nordic) who funded the studies directly. Studies were performed under quality systems at three U.S. ABLS4 labs, using the same parent strain for the preparation of a well-characterized challenge stock, EBOV 9510621. The EBOV challenge was carried out via three different routes of administration—intramuscular (IM), intranasal (IN), and aerosol exposure—using a target dose of 0.1 to 100,000 pfu. Overall, the meta-analysis described the post-EBOV challenge phase of the vaccine studies and demonstrated the consistency of the model at three ABSL4 sites. In lieu of a complete set of natural history studies at each site, the NHP control meta-analysis and supporting dataset was submitted to the FDA as a master file to the Center for Biologics Evaluation and Research (CBER). This meta-analysis and dataset may be leveraged by product developers and referenced in their Biologics License Application (BLA) to support vaccine licensure.

## 4. Discussion and Summary

There are several advantages to the strategy of combining data from challenged, untreated CMs from multiple studies. A major benefit was the reduction in the number of NHPs required for natural history studies at each ABSL4 site. We support the principles of the “3Rs”: to reduce, refine, and replace animal use in testing when feasible. Reduction includes methods which allow the maximum utilization of data and data sharing between groups and organizations to reduce the number of animals involved in studies. This goal was clearly accomplished in the filovirus meta-analysis. Another advantage of the strategy is the ability to demonstrate cross-laboratory applicability. This can reduce demand for ABSL4 space, which in turn increases accessibility. Increased data sharing and increased accessibility may aid in the identification of true disease pathogenesis outliers and disease markers of clinical significance. As product development advances, associated non-clinical studies using these model(s) and assays can target the most clinically relevant questions. Reduced numbers of animals and studies, as well as the more efficient utilization of ABSL4 space, should reduce product development costs and timelines. 

This approach may also be applied to existing NHP data from MARV and SUDV vaccine efficacy studies. Although multiple NHP efficacy studies have been conducted, adequate and well-controlled natural history studies for MARV and SUDV are still lacking at many of the ABSL4 sites. A meta-analysis of the existing NHP data may demonstrate the consistency of the model across the ABSL4 sites, which would support the development of vaccine products through the AR pathway. 

## 5. Conclusions

In conclusion, this successful strategy is an example of how a public–private partnership can leverage animal model data to support vaccine licensure under the AR pathway. This approach has also been applied to NHP Rhesus studies for the Zika virus (unpublished data, NIAID), where a meta-analysis of control animal data has been submitted to the FDA’s CBER in a master file to support vaccine regulatory submission and the licensure of products. Furthermore, this successful approach may be applied to support vaccine development for SUDV, MARV, and other pathogens of high consequence, and could be applied to the development of therapeutics through the FDA’s AR. 

An overarching lesson from the Ebolavirus outbreak is the importance of preparedness. The availability of animal models for biodefense facilitates the timely development of vaccines in response to a public health emergency. This example underscores the utility of mining existing data across many studies as an alternative to conducting formal natural history studies that will be increasingly difficult to perform owing to constraints on the supply of NHPs. Thus, it is important to have product development tools available that can be leveraged in case of an event, outbreak, or public health emergency.

## Data Availability

Not applicable.

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
