# Peer review of "A U.S. Government-Coordinated Effort to Leverage Non-Human Primate Data to Facilitate Ebolavirus Vaccine Development"

_vaccines, 2022, doi:10.3390/vaccines10081201_

Round 1
Reviewer 1 Report
This is a well written brief review describing a unique example of public-private partnership to generate non-human primates control data to facilitate approval the first Ebola vaccine using the FDA’s Animal Rule regulatory pathway.
Under this rule, efficacy is established in animal models rather than in large scale clinical efficacy trials (Phase 2 and Phase 3).
Natural history of Ebola-Zaire in cynomolgous macaque model was studied in limited numbers of BSL-4 facilities. It was crucially important to collect all available data to ensure rigor of study control.
The authors described US Government coordinated efforts to develop standardized tools to support licensure of rVSV-based Ebola vaccine using a non-traditional pathway. Since 2011, when Filovirus Animal Non-Clinical Group (FANG) was established, the FANG established standardized tools and reagents for NHP model and immunogenicity assays to facilitate approval Ebola vaccines, ERVEBO and Zabdeno/Mvabea.
The authors reasonably concluded that FANG activities reduced, refined, and replaced NHPs use in efficacy studies based on AR regulation. This experience and strategy described in the review can be used as an example of successful strategy to support vaccine and other biologics licensure under the AR regulation.
This review will be helpful for virologists, vaccine developers and regulation authorities involved in the development of biologics against emerging viral infectious diseases.
Reviewer 2 Report
In this review “A US Government Coordinated Effort to Leverage Non-human Primate Data to Facilitate Ebolavirus Vaccine Development” by Kimberly L. Taylor and colleagues is an outline of how standardized product development tools and existing NHP control data set were used to help development of various vaccines for EBOV. In general, this is an important review, and the approach may be used in support of other MCM development. The review makes very broad statements such using tools and does not provide examples of specific cases such as how centralized virus accelerated vaccine development and approval. The authors must spend a bit more time in the last paragraph of the review. The reviewer does not see a connection between the concluding paragraph and the title. The title does not represent the content of the review. Below, see additional comments on the title. I have made several comments to further elevate readability of the review.
1) The title should be tweaked to encompass the points that were brought up in the review such as standardized methodologies, cells, virus, etc to support development of the two vaccines. Implications of having standardized methods for future outbreaks should be part of the title. Alternatively, the authors must focus the review on standardization of NHP data and use.
2) Throughout the review, the authors need to better define how these methods helped to license the two EBOV vaccines.
3) The authors concentrated on vaccines and barely mentioned how these tools supported development of the two licensed anti EBOV Abs.
4) The reviewer recognizes that the authors attempted to stay focus on vaccines (see comment #5 also). However, several therapeutics were also licensed during the EBOV outbreak. The NHP models may have exaggerated the efficacy of some of these therapeutics. The authors should take advantage of the contrast and suggests ways to further improve the models.
5) The authors did not mention therapeutics such as ST-246 or Abs against various toxins which were licensed using the AR. These MCM were licensed without support of FANG. What was the lesson learned from these approved MCM vs the FANG approach?
6) Lines 55-56 the authors mention “As a result, the ERVEBO® and Zabdeno/Mvabea vaccines followed different pathways to licensure, and their lessons learned are important to consider for the development of Sudan and Marburg vaccines.” Although the authors make a few sentences in lines 207-210 about using the FANG and animal models for MARV and SUDV, the manuscript lacks clarity on how MCM development for these two viruses can follow the EBOV path.
7) Lines 112-113 the authors state “However, for Zaire ebolavirus only a limited number of adequately designed and well-controlled natural history studies have been conducted for the different filoviruses.” This sentence needs to be rewritten.
8) The cynomolgus macaque (CM) model was recommended as the NHP model for MCM development. Although the authors write a few sentences on CM model (lines 141-150) but there is little to no discussion about the selection criteria for this model. What was so unique about CM vs rhesus or other NHP models. This needs further discussion.
9) Lines 213-214 the authors state “The availability of standardized tools available for biodefense was critical for timely development and licensure of MCMs in response to a public health emergency.” This sentence needs to be rewritten for clarity.
10) Lines 214-219 the authors state “More recently, the value of preparedness was shown in response to the COVID-19 pandemic; vaccine strategies or tools from SARS-CoV-1 and MERS-CoV were rapidly employed to develop tools and vaccines against SARS-CoV-2. This example underscores the importance having product development tools available that can be leveraged in case of an event, outbreak or public health emergency” The authors are attempting to build a bridge between availability of tools and models for SARS-CoV-1 and MERS and how these tools may have helped the response to SARS-CoV-2 pandemic. The authors need to expand on specific tools or models that accelerated the production of vaccines or therapeutics against SARS-CoV-2. It is not clear to the reviewer how the proposed model may have improved SARS-CoV-2 vaccine testing or development. Further, the authors should extend this discussion to future outbreaks.
Round 2
Reviewer 2 Report
The authors responded positively to the reviewer comments. I have no additional concerns. This manuscript can move forward.